# Examining the bacterial diversity including extracellular vesicles in air and soil: implications for human health

Hyunjun Yun[1☯], Ji Hoon Seo[2☯], Yoon-Keun Kim[3], Jinho Yang[4]*

1 The AI Convergence Appliances Research Center, Korea Electronics Technology Institute, Gwangju, Republic of Korea, 2 Department of Environmental Health, Korea University, Seoul, Republic of Korea, 3 MD Healthcare Inc., Seoul, Republic of Korea, 4 Department of Occupational Health and Safety, Semyung University, Jecheon-si, Chungcheongbuk-do, Republic of Korea

☯ These authors contributed equally to this work.
* iamjinho@semyung.ac.kr

## Abstract

As the significance of human health continues to rise, the microbiome has shifted its focus from microbial composition to the functional roles it plays. In parallel, interest in ultrafine particles associated with clinically important impact has been increasing. Bacterial extracellular vesicles (BEVs), involved in systemic microbiome activity, are nano-sized spherical vesicles (20 - 100 nm in diameter) containing DNA, RNA, proteins, and lipids. They are known to be absorbed into the body potentially through air and soil, circulate in the blood, and directly impact diseases by affecting organs. Therefore, the aim of this study is to examine the biodiversity of bacteria and BEVs and predicted functional pathways. We sampled air and soil samples in Seoul, Korea and analyzed metagenomics based on 16S rRNA sequencing. At the phylum levels, Firmicutes in BEVs from soil and air were significantly higher than in bacteria, and Acidobacteria in both bacteria and BEVs from soil were significantly higher than from air ($p < 0.05$). The most dominant genera were *Pseudomonas* in bacteria from air and soil; and *Escherichia–Shigella* in BEVs from air and soil. In addition, Two-component system (ko02020) and ATP-binding cassette transporters (ko02010) were dominant functional pathways in both air and soil. The most functional pathways and orthologous groups were significantly different between air and soil ($p < 0.05$). In conclusion, human health can be affected differently depending on type of environment. Future study is necessary to have a better understanding of human health effects from environmental microbiota.

## 1. Introduction

There are numerous microorganisms in the air and soil. Soil supports the richest variety of biological life on Earth, providing a habitat for numerous microorganisms, whereas the atmosphere is less conducive to their survival [1,2]. However, many airborne microbes are metabolically active [3,4]. Both metabolically active and inactive microbes affect human health; for example, several airborne bacteria are associated to negative health effects, including respiratory and infectious diseases, acute toxic effects, allergies, and even cancer [5]. In

**Data availability statement:** All relevant data are within the paper and its Supporting Information files.

**Funding:** This research was supported by the Basic Science Research Program through the National Research Foundation of Korea (NRF), funded by the Ministry of Education (RS-2023-00244833). The funders had no role in study design, data collection and analysis, decision to publish, or preparation of the manuscript.

**Competing interests:** The authors declare no conflicts of interest.

addition, soil biodiversity is associated with human health, such as ecological complexity and robustness [6]. Humans are exposed to environmental microbiomes through various routes, including ingestion and inhalation, as these microbiomes circulate among soil, water, and air. Consequently, the assessment of environmental microbiomes is crucial for understanding their impact on human health.

Recently, interest in the microbiome has shifted from composition to functional roles in human health. Furthermore, public concern for ultrafine particles, including bioaerosols, has increased due to their ability to remain airborne for extended periods and penetrate deep into the respiratory tract. Among these bioaerosols, bacterial extracellular vesicles (BEVs) are emerging as key factors in airborne microbial exposure, potentially influencing immune responses and respiratory health [7,8]. Understanding the differences between airborne and soil microbiomes, including BEVs, is critical for assessing their environmental and health implications. Ultrafine particles, when inhaled, rapidly distribute to various organs, including the lungs, liver, kidneys, and intestines. These particles penetrate tissue and persist within lung cells, suggesting a prolonged impact of particle accumulation in deep lung tissue on lung function over a period of four weeks [9]. Extracellular vesicles (EVs), a type of ultrafine particle, are produced as a result of systemic cellular activity. The biogenesis of EVs is a highly regulated process controlled by various signaling molecules and initiated by receptor activation specific to each cell type [10]. The biogenesis of eukaryotic EVs has been well characterized, with eukaryotic cell-derived EVs being classified into exosomes, ectosomes (or shedding vesicles), and apoptotic bodies [11]. In contrast, the biogenesis of prokaryotic cell-derived EVs has only recently been elucidated. Prokaryotic cell-derived EVs are categorized as ectosomes and apoptotic bodies [12]. BEVs are further classified into Gram-negative bacteria-derived outer membrane vesicles and Gram-positive bacteria-derived membrane vesicles. Previous studies have highlighted the role of BEVs in the pathogenesis of various diseases, including skin, lung, and gut diseases, metabolic disorders, central nervous system conditions, and cancers. For examples, *Staphylococcus aureus* EVs have been linked to the pathogenesis of atopic dermatitis, asthma, chronic obstructive pulmonary disease, chronic inflammatory airway diseases, and lung cancer. Additionally, *Escherichia coli* EVs was associated with emphysema, inflammatory bowel disease, and lung cancer [12]. Furthermore, *Pseudomonas aeruginosa* EVs and *Paenalcaligenes hominis* EVs have been related to pulmonary inflammation and cognitive impairment in vivo tests, respectively [13,14]. BEVs are nano-sized (20-100 nm in diameter), spherical, and composed of a lipid bilayer, which contributes to their high stability [15]. In addition, BEVs, including DNA, RNA, proteins, and lipids, play roles in intercellular transport and communication, as they deliver microbial components [16–18]. Some studies show that BEVs are absorbed into the body and can circulate in blood and affect the body's organs [19]. Overall, focus has increased regarding the influence of BEVs on health and disease, and many BEV studies have demonstrated that they can directly affect disease [20,21].

Culture-based methods in microbiology have been used to analyze microbial communities, enabling the direct observation of bacteria and fungi. In particular, airborne bacteria and fungi are monitored based on colony-forming unit (CFU) concentrations, as regulated by the World Health Organization using culture-based methods. Culture-based studies have shown that most airborne bacteria are Gram-positive bacteria (*Staphylococcus, Corynebacterium, Bacillus,* and *Micrococcus*), although some Gram-negative bacteria (such as *Acinetobacter, Moraxella, Pantoea,* and *Pseudomonas*) have been analyzed at lower frequencies [22–28]. In addition, culture-based methods cannot detect BEVs, as BEVs cannot be cultured. The composition of airborne microbial communities differs between culture-based methods and 16S rRNA gene sequencing [29,30]. The total 16S DNA yield from a sample can include approximately a quarter of that derived from BEVs [31].

Therefore, research on environmental applications of BEVs is crucial. BEVs hold significant potential for various environmental applications, including serving as bioremediation agents in environmental disaster mitigation, removing problematic biofilms, and aiding in waste treatment. Further investigation in this field is essential to fully understand the environmental and ecological significance of BEVs and to explore how they can be effectively utilized as tools in a wide range of environmental contexts [32].

While previous studies have examined airborne and soil microbiomes separately, there has been no in-depth study comparing bacterial communities and BEVs between these two environments. Furthermore, little is known about the functional roles of BEVs in environmental microbiomes and their potential implications for human health. Therefore, this study aims to provide a comprehensive analysis of bacterial and BEV microbiota in air and soil using 16S rRNA sequencing, with a specific focus on their biodiversity and predicted functional pathways. We hypothesize that the bacterial community composition and functional pathways of BEVs differ significantly between air and soil, which may influence human health in distinct ways.

## 2. Materials and methods

### 2.1. Sampling

Air samples for airborne microorganisms were collected in sterilized Petri dishes using the gravitational settling method for a day per sample [33]. Ambient air sampling was conducted on the rooftop of a building during a period without precipitation or strong winds, after obtaining access permission. Simultaneously, soil samples were gathered in sterilized tubes from the surface near a mountain path close to the building. Sampling was conducted over 20 sunny days from May to July 2016 in Seoul, South Korea (37.58° N, 126.88° E), during which there were no significant events, such as the rainy season or yellow dust occurrences. In total, 80 samples were collected, consisting of 20 airborne bacteria (AB) samples, 20 airborne bacterial extracellular vesicles (AE) samples, 20 soil bacteria (SB) samples, and 20 soil bacterial extracellular vesicles (SE) samples. All samples were stored below -20 °C.

### 2.2. DNA isolation and sequencing

Air samples in a Petri dish and 5 g soil samples were each diluted in 10 mL of phosphate-buffered saline (PBS) for 24 hours at 4°C, vortexed, and filtered through a cell strainer. Additionally, two PBS samples were analyzed as blanks (negative controls), and a standard material was used to monitor for errors, such as contamination and excessive growth. The pellet and supernatant—containing cells and EVs, respectively—were separated via centrifugation at 10,000 x g for 10 minutes at 4 °C. To eliminate cell and foreign particles in the supernatant, filtering was performed using a 0.22 um filter. After the filtered samples were boiled and centrifuged at 13,000 rpm at 4 °C, DNA was extracted from the pellets and supernatant using a DNeasy PowerSoil kit (QIAGEN, Germany) [34,35]. Therefore, DNA in the pellets represents bacteria, while DNA in the supernatant indicates BEVs. In this study, we employed 16S rRNA-based metagenomic analysis to characterize bacterial communities rather than whole macrogenome sequencing. This approach was chosen to efficiently compare bacterial and BEV compositions in airborne and soil environments. Bacterial genomic DNA targeting the V3–V4 hypervariable regions of the 16S rRNA gene was amplified with the 16s_V3_F and 16s_V4_R primers [34,35]. Library preparation was conducted using PCR products, which were quantified with QIAxpert (QIAGEN, Germany), and each amplicon was sequenced using MiSeq (Illumina, USA). Raw pyrosequencing reads obtained from the sequencer were filtered based on barcode and primer sequences using MiSeq technology (Illumina, USA).

### 2.3. Taxonomic assignment and functional pathway profiles.

Taxonomic assignment was performed based on SILVA database 128 with UCLUST. This program uses Cutadapt (ver. 1.1.6) for trimming, CASPER for merging, and VSEARCH for clustering into operational taxonomic units (OTUs) as previously described [34,35]. To ensure high-quality sequencing reads, sequences with read lengths shorter than 350 bp or longer than 550 bp, as well as those with Phred quality scores below 20, were discarded [36]. During this process, subsampling was conducted based on the sample with the lowest read count among all samples to ensure comparability between them. Taxonomic assignments were made based on sequence similarities, with thresholds set as follows: species level (>97% similarity), genus level (>94% similarity), family level (>90% similarity), order level (>85% similarity), class level (>80% similarity), and phylum level (>75% similarity). OTUs with fewer than 10 sequences in only one sample were excluded from further analysis. To predict the possible functional pathway, we used Tax4Fun [37] to calculate the contributions of various OTUs to known biological pathways based on KEGG orthology groups (KOs) using Kyoto Encyclopedia of Genes and Genomes (KEGG) databases [38].

### 2.4. Statistical analysis

Significant differences in microbiome, functional pathway, and orthologous groups between the sample types were assessed using a Mann–Whitney test for continuous variables. Findings were considered significant for $p < 0.05$. The alpha diversity of the microbial composition was measured using the observed OTUs, Chao1, Shannon index, and Simpson index and rarified to compare species richness and evenness. To adjust for uneven sequencing depth among the samples, the samples were rarefied to even depths of 10,000 before diversity analysis. Beta diversity (diversity among samples within a group) was calculated based on the Bray-Curtis dissimilarity index and visualized using Principal Coordinate Analysis (PCoA). The p-value for PCoA was calculated via Permutational Multivariate Analysis of Variance (PERMANOVA) using distance matrices. Correlations among microbiome, functional pathway, and orthologous groups were analyzed using Pearson correlation coefficients. All statistical analyses were carried out using R version 4.3.2.

## 3. Results

### 3.1. Difference in diversity between air and soil

We obtained a total of 3,000,193 high-quality sequences. The average number of valid reads was 33,974.1 ± 12,211.2 for AB, 42,293.5 ± 28,041.7 for AE, 43,822.1 ± 26,526.2 for SB, and 41,829.3 ± 23,282.8 for SE.

To analyze the alpha diversity for richness and evenness, Chao1, observed OTUs, the Shannon index, and the Simpson index were analyzed. These alpha diversity indexes were significantly higher for SB than the other sample types ($p < 0.05$) (Fig. 1A). However, there was no significant difference in alpha diversity between AB, AE, and SE. PCoA, representing beta diversity, yielded significantly distinct clustering between air and soil ($p < 0.05$). Additionally, differences were observed between bacteria and BEVs in air and soil samples (Fig. 1B).

### 3.2. Metagenomic analysis of bacteria and bacterial extracellular vesicles in air and soil

At the phylum level, Proteobacteria were the most dominant in every sample type, with 74.9 ± 20.6% in AB, 40.4 ± 21.4% in AE, 44.1 ± 20.4% in SB, and 47.5 ± 12.1% in SE.

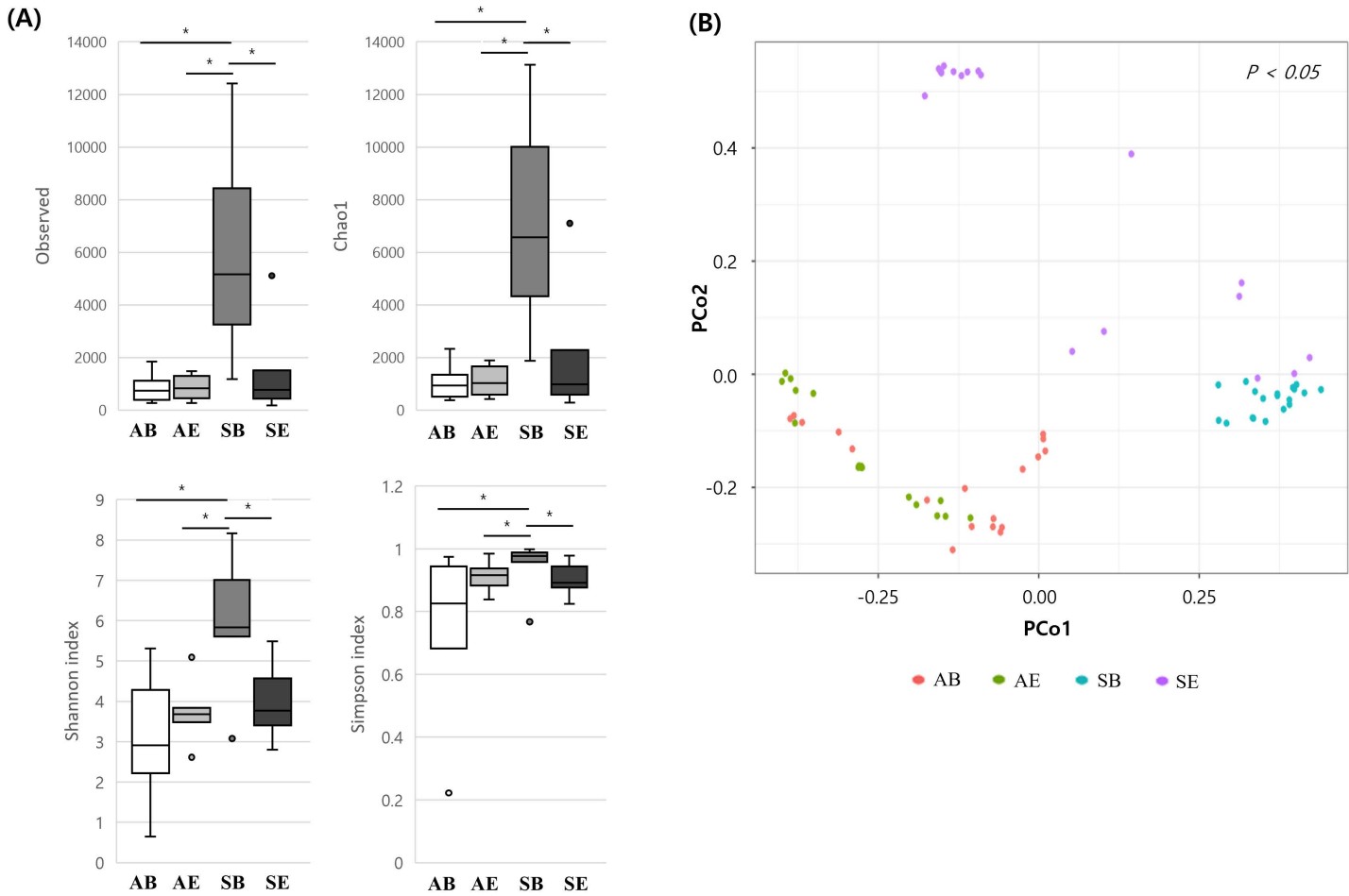

**Fig. 1. Differences in alpha and beta diversity between airborne bacteria (AB), airborne BEVs (AE), soil bacteria (SB), and soil BEVs (SE).** (A) Alpha diversity: observed OTUs, Chao1, Shannon index, and Simpson index. (B) Beta diversity: PCoA of the bacteria and BEVs. (*: $p < 0.05$.).

Firmicutes, Bacteroidetes, and Actinobacteria were also among the next most abundant phyla in every sample type. Firmicutes were the second most abundant in AB, AE, and SE, while Bacteroidetes were the second most abundant in SB. In addition, Firmicutes were significantly more abundant in BEVs from both air and soil than in bacteria ($p < 0.05$). Acidobacteria in soil (9.9 ± 8.5% in SB and 3.3 ± 5.7% in SE) were significantly higher than in air (0.2 ± 0.3% in AB and 0.0 ± 0.2% in AE) ($p < 0.05$) (Fig. 2A, Supplementary Table S1). At the genus levels, the most dominant genera were *Pseudomonas* in AB at 12.6 ± 28.4% (62,355 reads) and SB at 18.1 ± 16.9% (141,722 reads); and *Escherichia–Shigella* in AE at 14.5 ± 10.9% (139,158 reads) and SE at 19.5 ± 14.0% (176,979 reads). However, the next most dominant genera in SB and SE differed based on the average relative abundance and the total read counts (Fig. 2B, 2C, Supplementary Table S1). In addition, there was a difference between air and soil; *Methylobacterium* in air was significantly higher than in soil, while *Arthrobacter* in air was significantly lower than in soil ($p < 0.05$). Furthermore, *Escherichia–Shigella, Proteus, Lactobacillus*, and *Enterococcus* in bacteria were significantly lower than in BEVs from air and soil ($p < 0.05$) (Supplementary Table S1).

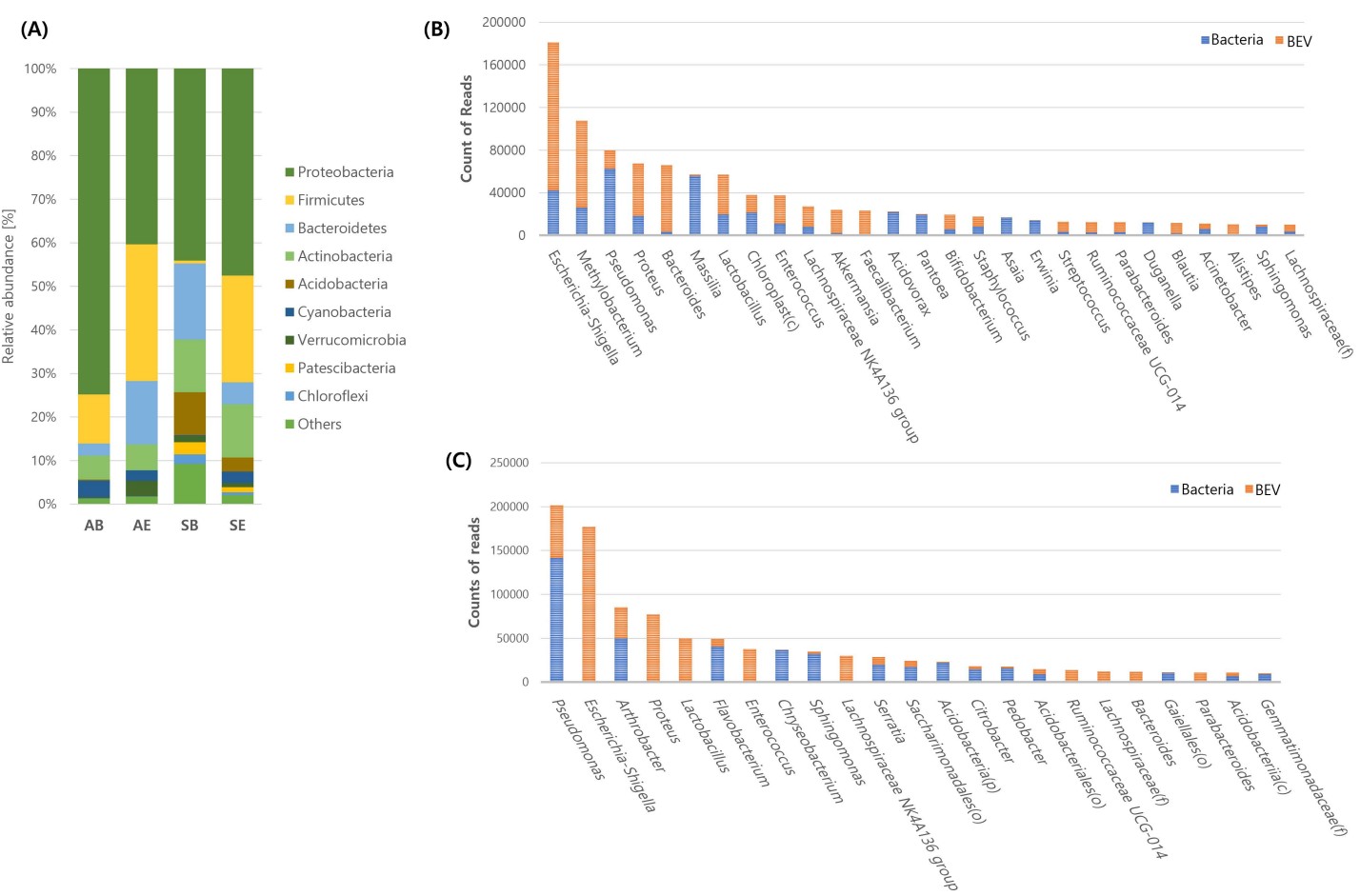

**Fig. 2. Dominant microbiome in air and soil.** (A) Averaged microbiota at the phylum level. (B) Dominant genera of bacteria and BEVs based on total read counts in air samples. (C) Dominant genera of bacteria and BEVs based on total read counts in soil samples.

### 3.3. Functional pathway and orthologous groups

The dominant functional pathways in air samples were the ATP-binding cassette (ABC) transporters (ko02010) and the two-component system (ko02020) (Supplementary Fig. S1A). In addition, the dominant functional pathway in soil was the metabolic pathway (ko01100), followed by microbial metabolism in diverse environments (ko01120), the biosynthesis of secondary metabolites (ko01110), the biosynthesis of antibiotics (ko01130), ABC transporters (ko02010), and the two-component system (ko02020) (Supplementary Fig. S1B). The most functional pathways were significantly different between air and soil (p < 0.05), while the metabolic pathway (ko01100), microbial metabolism in diverse environments (ko01120), biosynthesis of secondary metabolites (ko01110), the biosynthesis of antibiotics (ko01130), and ABC transporters (ko02010) were not significantly different between bacteria and BEVs (Fig. 3A).

In the orthologous groups, the dominant groups in the air included the iron complex outer-membrane receptor protein (K02014); the ATP-binding cassette, subfamily B, bacteria (K06147); the methyl-accepting chemotaxis protein (K03406); hydrophobic/amphiphilic exporter-1 (mainly G- bacteria); and the HAE1 family (K03296) (Supplementary Fig. S2A). In addition, the dominant orthologous groups in the soil were 3-oxoacyl-[acyl-carrier protein] reductase (K00059) and ATP-binding protein in several systems (K02031, K02003, K02032,

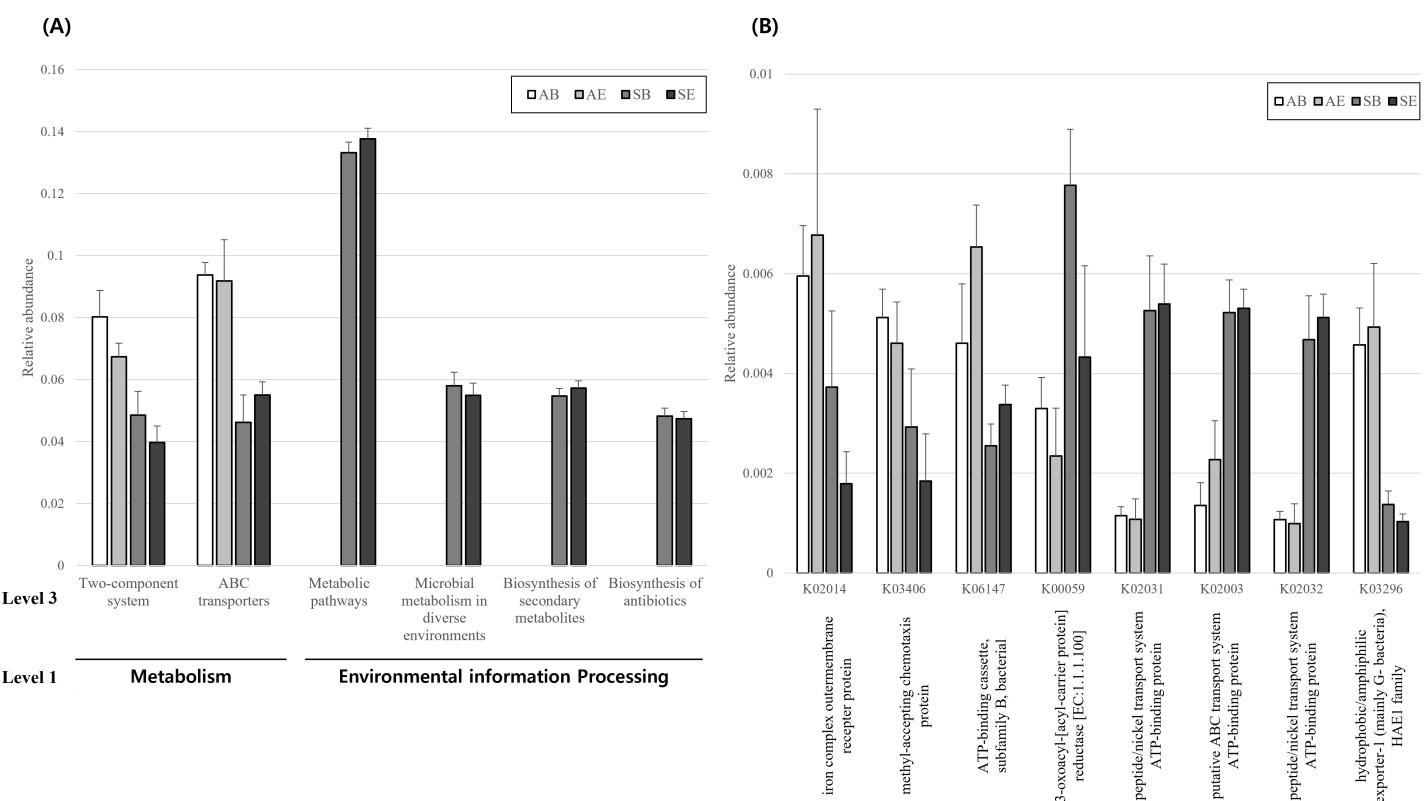

**Fig. 3. Difference in dominant (A) functional pathway and (B) orthologous groups between airborne bacteria (AB), airborne BEVs (AE), soil bacteria (SB), and soil BEVs (SE).**

K01990, K02049, K02028, K02013, K02010, K02052, K01995, K10112, and K01996) (Supplementary Fig. S2B). The most orthologous groups were significantly different between the air and soil ($p < 0.05$), whereas hydrophobic–amphiphilic exporter-1 (mainly G- bacteria), the HAE1 family (K03296), the putative ABC transport system ATP-binding protein (K02003), the iron complex outer-membrane receptor protein (K02014), the peptide–nickel transport system ATP-binding protein (K02031), and the peptide–nickel transport system ATP-binding protein (K02032) were not significantly different between bacteria and BEVs (Fig. 3B).

## 3.4. Genera correlated with functional pathway and orthologous groups

*Escherichia–Shigella* in AB was highly correlated with dominant functional pathways and orthologous groups. In particular, correlation coefficients with Purine metabolism (ko00230) and Aminoacyl-tRNA biosynthesis (ko00970) of the functional pathway were high ($r > 0.9$, $p < 0.01$), while correlation with the iron complex outer-membrane receptor protein (K02014) and the HAE1 family (K03296) were extremely negative ($r = -0.95$ and $-0.93$, respectively, $p < 0.05$) (Fig. 4A). *Bacteroides* in AE showed significant positive correlation with Pyrimidine metabolism (ko00240) in functional pathway and the iron complex outer-membrane receptor protein (K02014) and the putative ABC transport system permease protein (K02004) in the orthologous group ($r > 0.9$, $p < 0.01$) (Fig. 4B). In SB, opposite correlations with dominant functional pathways were found between *Pseudomonas* and *Arthrobacter*. In addition, *Pseudomonas, Chryseobacterium*, and *Flavobacterium* in SB were significantly negatively associated with microbial metabolism in diverse environments (ko01120) ($r < -0.5$, $p < 0.05$) (Fig. 4C).

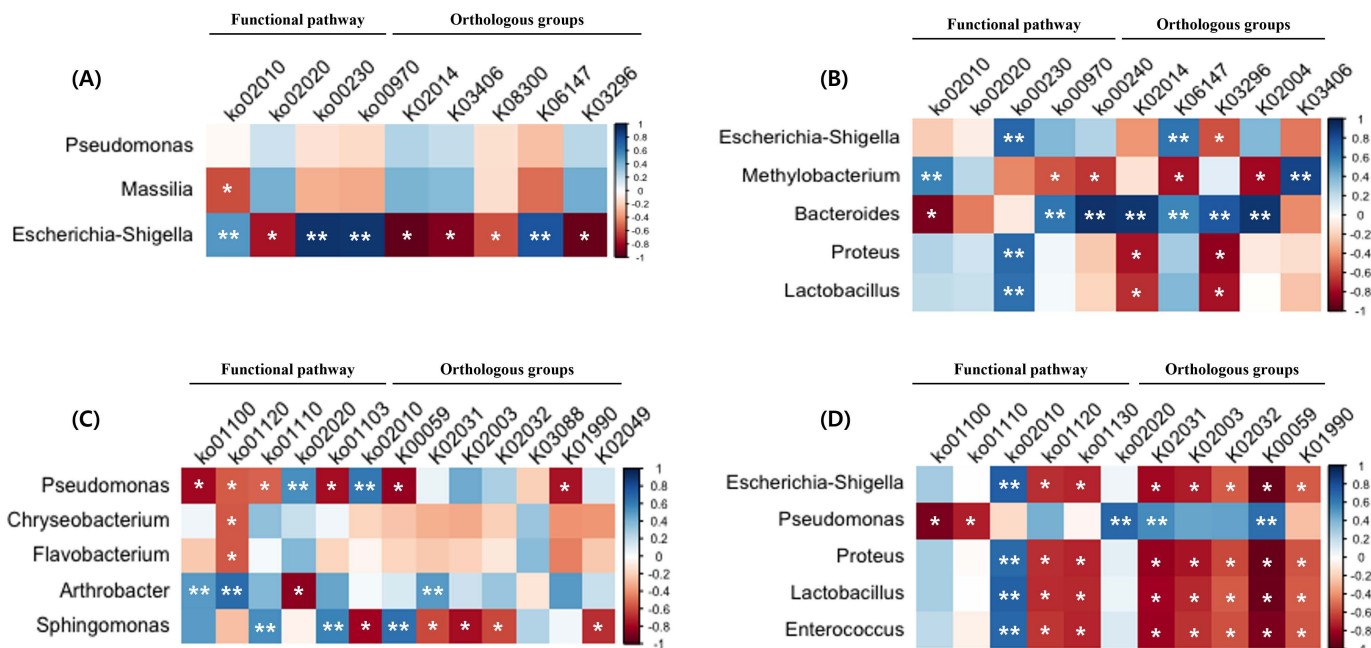

**Fig. 4. Heatmap of correlation between dominant genera, dominant functional pathways, and orthologous groups in (A) airborne bacteria, (B) airborne BEVs, (C) soil bacteria, and (D) soil BEVs.** Color is based on Pearson correlation coefficients.

*Escherichia–Shigella, Proteus, Lactobacillus,* and *Enterococcus* in SE were significantly negatively correlated with dominant orthologous groups and two functional pathways, including microbial metabolism in diverse environments (ko01120) and biosynthesis of antibiotics (ko01130) (p < 0.05). Furthermore, *Pseudomonas* in SE exhibited a significant negative correlation with the metabolic pathway (ko01100) (r = -0.89, p < 0.05) (Fig. 4D).

## 4. Discussion

In this study, significant differences in bacterial and BEV composition were observed between air and soil samples. *Pseudomonas, Massilia,* and *Escherichia–Shigella* were dominant in airborne bacteria, while *Escherichia–Shigella, Methylobacterium,* and *Bacteroides* were the most abundant in airborne BEVs. Comparisons with previous studies indicate that microbial community composition can vary based on environmental conditions, including geographical location, season, and methodology. While previous research has identified *Methylobacterium, Streptomyces, Pseudonocardia, Sphingomonas,* and *Bacillus* as dominant airborne bacteria [33,39], our findings highlight the need to further investigate the factors shaping microbial distributions in different environments. In soil, the dominant genera were *Pseudomonas, Corynebacterium*, and *Flavobacterium* in bacteria and *Escherichia–Shigella, Pseudomonas,* and *Proteus* in BEVs, while *Sphingomonadaceae(f), Xanthomonadaceae, Comamonadaceae(f),* and *Sphingomonas* were the dominant genera in control soil in a previous study [40]. In addition, Proteobacteria, Actinobacteria, and Acidobacteria are the dominant phyla in Mediterranean mountains [41]. Variability in microbial community composition can be attributed to differences in methodology, environmental conditions, and sample processing techniques. One critical factor is the separation of bacteria and BEVs, as genomic DNA extractions typically contain both bacterial and BEV-derived DNA [31]. A methodology that can separate micro-sized bacteria and nano-sized BEVs is essential for clear microbiome analysis. In addition,

many environmental factors affect the soil microbiome. Acidobacteria in soil were the most abundant phylum in this study. The abundance of Acidobacteria in the low-altitude forest sample was higher than in some studies of altitudinal gradients [42,43]. In some cases, the abundance of Acidobacteria showed a negative correlation with soil pH [43] or a positive correlation with carbon inputs [44]. Soil microbial communities are strongly affected by pH, altitudinal gradient, and abiotic factors [45,46]. Therefore, we suggest that assessing environmental factors is necessary for managing microbial environments since environmental factors strongly influence microbiomes.

Both airborne and soil microbiomes play a significant role in human health, with airborne BEVs receiving increasing attention due to their potential to influence immune responses. The composition of BEVs in air and soil differs significantly, which may lead to distinct health effects based on exposure routes. For example, inhalation of BEVs from airborne bacteria, such as *S. aureus* EVs, has been associated with neutrophilic pulmonary inflammation and neutrophilic asthma [47,33]. Further research is required to elucidate the mechanisms by which BEVs interact with the human body and their potential long-term health effects. Additionally, exposure to *E. coli* BEVs has been linked to gut and lung inflammatory responses, suggesting a possible systemic impact. In contrast, soil-derived BEVs may enter the body through ingestion or dermal contact, potentially modulating immune responses in the gut microbiome. While some BEVs have been shown to have antitumor properties, such as their ability to induce an immune response against metastatic carcinoma in lung tissues [48–50], the exact mechanisms by which BEVs interact with human immune systems remain unclear. Future studies should investigate the immunomodulatory properties of environmental BEVs to determine their potential risks and benefits in different exposure scenarios. BEVs derived from indoor dust are also associated with pulmonary disease. In vivo testing has shown that BEVs in indoor dust can induce emphysema in the lungs [51,52]. *S. aureus* EVs and *E. coli* EV have been shown to trigger the Th17 immune response. Specifically, these EVs promote IL-17 production through the activation of polarized Th17 cells, leading to neutrophilic inflammation. This inflammatory response, characterized by neutrophil recruitment, can induce epithelial cell dysplasia and upregulate expression of matrix metalloproteinases, potentially contributing to the development of lung cancer [7]. Lung cancer is one of the most prevalent and fatal diseases globally, leading to a rise in prediction research, and BEVs have shown potential as a factor in lung cancer prediction and medicine [53–55]. These previous studies have shown that BEVs derived from environmental samples can elicit an immune response. Therefore, we suggest that it is important to analyze the BEV microbiome for human health, especially pulmonary and respiratory diseases.

In a previous study, oxidative phosphorylation (ko00190; Level2: energy metabolism) was the most functional pathway in mountain soil [41]; however, in this study, the metabolic pathway (ko01100; Level2: global and overview maps) was the most functional pathway. Interestingly, ABC transporters (ko02010) and two-component systems (ko02020) were the most common functional pathways analyzed in all groups, and they were significantly higher in the air groups than in the soil group. While this superfamily is ubiquitous in all domains of life, many ABC transporters are associated with bacterial pathogenicity, with contributions to toxins; protein and polysaccharide secretion; drug efflux functionality; and lipopolysaccharide biogenesis in Gram-negative bacteria [56]. Additionally, two-component signal transduction systems are widespread in prokaryotes and extensively mediate adaptive responses to environmental changes [57,58]. The ABC transporter is related to lung functions [59], and two-component systems in pathogens are concerned with sensing antibiotics and regulating the gene expression of antibiotic resistance in response to antibiotic exposure signals [60]. The arc two-component signal transduction system plays a role in adapting *E. coli* by changing the

respiratory conditions of growth [61]. In addition, the GacS/GacA two-component system critically affects the many beneficial and pathogenic communications of Gram-negative bacteria with their host organisms [62]. Our results indicate that functional pathways related to signaling, bacterial pathogenicity, and immune system interactions are more abundant in airborne microbiomes compared to soil. This suggests a potential link between environmental exposure and immune responses, particularly through inhalation. Further studies utilizing in vivo and in vitro models are necessary to establish the functional relevance of these pathways and their potential health implications.

The composition of the airborne microbiome is closely associated with aerosol sources, such as freshwater, cropland, soil, and urban environments [63]. Moreover, ambient bioaerosols can influence indoor bioaerosols, leading to human exposure through inhalation and skin contact. Consequently, we can be exposed to microbiomes present in any environment. The growth and death of bacteria in the environment, including soil, water, and air, are largely affected by environmental conditions, and in the same way, BEVs such as the ectosome and the apoptotic body are derived from the growth and death of bacteria [12]. Since bacteria and BEVs circulate between air and soil, the microbiota of air and soil can be affected by each other. However, there were differences in microbiota and functions between air and soil in this study. Although these influences are mutual, bioaerosols are influenced by more environmental factors, so it is not enough to judge the human health impact of bioaerosols based on environmental assessment. Additionally, there are studies on microorganisms based on CFUs, cultures, and sequencing, and studies based on new generation sequencing (NGS) and shotgun sequencing have recently increased. This study is the first to analyze the differences between air and soil microbiomes, especially regarding BEVs. We also analyzed the functional pathway and orthologous groups based on microbiomes in air and soil. Our results showed differences between the sample types. We suggest that health effects may be different depending on the type of environment we are exposed to. Therefore, management using CFU concentrations has limitations, and we propose that nano-sized particles, including BEVs, should be controlled. However, the technology to monitor nano-sized bioaerosols is currently insufficient, making developing new technology necessary [64]. Recently, filters capable of removing nano-sized particles have been developed to improve indoor air quality using air purifiers [65,66].

This study has several limitations. First, sampling was conducted during a single season at a limited number of locations, which may not fully capture the variability of microbial communities across different temporal and geographical scales. Expanding the dataset to include multiple seasons and diverse environmental conditions would provide a more comprehensive understanding of microbial dynamics. The number of samples was insufficient to characterize environmental microbiome. Second, we used a gravitational settling method for sampling, but this approach did not capture airborne microbiomes sufficiently, particularly BEVs. BEVs are nano-sized and do not readily sediment or flocculate, resulting in a longer retention time in the atmosphere [67,68]. Third, NGS has limitations in analyzing bacterial species owing to a lack of reference sequences [69]. Shotgun metagenome sequencing offers more comprehensive information compared to 16S rRNA sequencing and may therefore provide greater accuracy in microbial analysis [70]. Using shotgun sequencing would enable a more precise identification of species. However, shotgun sequencing presents challenges related to time and cost, which can limit its practical application. To assign taxonomy to sequences, researchers commonly use public databases such as GREENGENES, RDP, NCBI, Silva, and EzTaxon The choice of database can vary between studies, with each offering distinct differences in terms of diversity, accuracy, and update frequency. Finally, the methodologies for isolating BEVs

from environmental samples remain non-standardized. Variations in ultracentrifugation techniques—including differences in speed, duration, repetition, and the specific kits used—contribute to inconsistencies. Furthermore, there are currently no established quality assurance and quality control criteria for assessing BEV purity, such as size and quantity. These factors can lead to discrepancies in BEV data depending on the research methods employed. To address these issues, future studies should implement standardized methodologies and include samples from diverse sites.

## 5. Conclusions

This study reveals distinct differences in bacterial and BEV microbiomes between air and soil, emphasizing their potential influence on human health through varied exposure routes. Airborne BEVs may play a role in respiratory immune responses, while soil-derived BEVs could impact gut microbiota via ingestion or dermal contact. Given their biological significance, further research should investigate the functional roles of BEVs in immune modulation and disease mechanisms using in vitro and in vivo models. Expanding environmental sampling across different seasons and locations, along with standardized BEV isolation methods, will improve our understanding of their health impacts. Additionally, the development of BEV-based biomarkers could enhance environmental monitoring and public health risk assessments, contributing to improved air and soil quality management.

## Supporting information

**S1 Fig. Dominant functional pathway of bacteria and BEVs in (A) air and (B) soil.** (PPTX)

**S2 Fig. Dominant orthologous groups of bacteria and BEVs in (A) air and (B) soil.** (PPTX)

**S1 Table. Relative abundance of microbiome at the phylum and genus levels.** (XLSX)

**S1 File. Metagenomics raw data.** (XLSX)

## Acknowledgment

We would like to acknowledge Semyung University and MD Healthcare Inc. for supporting article processing.

## Author contributions

**Conceptualization:** Yoon-Keun Kim, Jinho Yang.

**Formal analysis:** Hyunjun Yun, Ji Hoon Seo, Jinho Yang.

**Funding acquisition:** Jinho Yang.

**Investigation:** Hyunjun Yun, Ji Hoon Seo.

**Methodology:** Ji Hoon Seo, Yoon-Keun Kim, Jinho Yang.

**Project administration:** Yoon-Keun Kim, Jinho Yang.

**Supervision:** Yoon-Keun Kim, Jinho Yang.

**Visualization:** Hyunjun Yun, Ji Hoon Seo, Jinho Yang.

**Writing – original draft:** Hyunjun Yun, Ji Hoon Seo.

**Writing – review & editing:** Yoon-Keun Kim, Jinho Yang.

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
