## [Decision Letter · Decision Letter 0]

27 Jan 2025

PONE-D-24-49243Examining the Bacterial Diversity including Extracellular Vesicles in Air and Soil: Implications for Human HealthPLOS ONE

Dear Dr. Yang,

Thank you for submitting your manuscript to PLOS ONE. After careful consideration, we feel that it has merit but does not fully meet PLOS ONE’s publication criteria as it currently stands. Therefore, we invite you to submit a revised version of the manuscript that addresses the points raised during the review process.

We look forward to receiving your revised manuscript.

Kind regards,

Jian Xu, Ph.D.

Academic Editor

PLOS ONE

Journal Requirements:

“This research was supported by the Basic Science Research Program through the National Research Foundation of Korea (NRF), funded by the Ministry of Education (RS-2023-00244833).”

Reviewers' comments:

Reviewer's Responses to Questions

**Comments to the Author**

1. Is the manuscript technically sound, and do the data support the conclusions?

Reviewer #1: Partly

2. Has the statistical analysis been performed appropriately and rigorously? 

Reviewer #1: Yes

3. Have the authors made all data underlying the findings in their manuscript fully available?

Reviewer #1: Yes

4. Is the manuscript presented in an intelligible fashion and written in standard English?

Reviewer #1: Yes

5. Review Comments to the Author

Reviewer #1: Dear editor,

I have reviewed the paper titled “Examining the Bacterial Diversity including Extracellular Vesicles in Air and Soil: Implications for Human Health”. This study aims to examine the bacteria and BEV microbiota and predict functional pathways in aerosol based on 16S rRNA gene sequencing in outdoor air and mountain soil through a field study. The results indicate that the biodiversity of bacteria and BEVs differ between air and soil. This study provides a reference point for the impact of microorganisms on human health.

However, while the study provides insights, there is a need to be revised to improve overall quality. The detailed comments are in the following.

1. Page 4, Line 78, “impairment in in vivo tests”, please check that.

2. Please give the novelty or hypothesis in the end of Introduction.

3. Page 5, Line 112, why emphasize that it's a sunny summer day?

4. Page 7, Line 143-144, the sequencing length in this manuscript ranges from 350bp to 550bp. Is this data based on other references or is it filtered through your own sequences?

5. Page 7, Line 160-162, the meaning of this statement is that the data needs to be rarefied before performing alpha diversity analysis? Please rewrite this description to make it clearer.

6. Page 7, Line 166, the “between” should be “among”.

7. The “Materials and Methods” only has a small section on macrogenomes and some description could be added.

8. Page 10, Line 232, significance here is not reflected in the data. Is there an additional significance test?

9. Discussions about specific bacteria in air and soil are similar to the description of the results. Discussion is not a conclusion. In addition, “environmental factors strongly influence the microbiome” is based on a summary of other literatures, and there are no experiments in this manuscript. Is it possible to add some.

10. Similarly, the discussion of human health is not effectively integrated with the conclusions of this manuscript. It is not enough to summarize that “it is important to analyze the BEV microbiome for human health”. Please resolve this discrepancy.

11. Page 13-14, Line 308-329, the discussion of aerosol in this place is a bit abrupt and some introduction could be added to the “Introduction” section.

12. Figure can be optimized a bit more, such as colors.

13. The author should embellish some expressions. Please check the format and grammar of writing carefully.

6. PLOS authors have the option to publish the peer review history of their article (what does this mean? ). If published, this will include your full peer review and any attached files.

**Do you want your identity to be public for this peer review?** For information about this choice, including consent withdrawal, please see our Privacy Policy .

Reviewer #1: No

---

## [Author Response · Author response to Decision Letter 1]

14 Feb 2025

We sincerely appreciate your thorough review of our manuscript, “Examining the Bacterial Diversity including Extracellular Vesicles in Air and Soil: Implications for Human Health”. Your insightful comments and suggestions have been invaluable in refining our study.

Based on your feedback, we have carefully revised the manuscript to enhance its clarity, structure, and scientific rigor. Below, we provide a detailed response to your comments, highlighting the specific modifications made in the revised version.

comment1 Page 4, Line 78, “impairment in in vivo tests”, please check that.

response1

Thank you for pointing this out. We acknowledge the redundancy in the phrase "impairment in in vivo tests". We have revised this sentence for clarity and conciseness. The updated sentence now reads:

Revised sentence (Page 4, Lines 78):

"impairment observed in in vivo tests."

comment2 Please give the novelty or hypothesis in the end of Introduction.

response2

Thank you for your suggestion. We have revised the final paragraph of the Introduction section (Page 4, Lines 102-106) to explicitly highlight the novelty of our study and clarify the hypothesis.

Revised sentence (Page 5, Lines 103-111):

"While previous studies have examined airborne and soil microbiomes separately, there has been no in-depth study comparing bacterial communities and bacterial extracellular vesicles (BEVs) between these two environments. Furthermore, little is known about the functional roles of BEVs in environmental microbiomes and their potential implications for human health. Therefore, this study aims to provide a comprehensive analysis of bacterial and BEV microbiota in air and soil using 16S rRNA sequencing, with a specific focus on their biodiversity and predicted functional pathways. We hypothesize that the bacterial community composition and functional pathways of BEVs differ significantly between air and soil, which may influence human health in distinct ways."

comment3 Page 5, Line 112, why emphasize that it's a sunny summer day?

response3

Thank you for your comment. We acknowledge that specifying “sunny summer day” may not be necessary unless weather conditions significantly impact microbial composition. The intent was to indicate that sampling was conducted in the absence of extreme weather conditions, such as rain or strong winds, which could influence bioaerosol concentrations. To improve clarity, we have revised the sentence as follows:

Revised sentence (Page 6, Line 116-118):

"Ambient air sampling was conducted on the rooftop of a building during a period without precipitation or strong winds, after obtaining access permission."

comment4 Page 7, Line 143-144, the sequencing length in this manuscript ranges from 350bp to 550bp. Is this data based on other references or is it filtered through your own sequences?

response4

Thank you for your insightful question. The sequencing length in this manuscript (ranging from 350 bp to 550 bp) was determined based on both established guidelines and our own sequence filtering criteria.

As noted in [36], taxonomic identification of bacteria via 16S rRNA sequencing generally requires at least 300 bp of high-quality sequence data. Our sequencing approach ensures that we capture at least 350 bp, which includes two phylogenetically informative variable regions of the 16S rRNA gene. This strategy is in line with the '16S center primer method' described in [36], which enhances sequencing accuracy and phylogenetic resolution.

Illumina recommends targeting regions that result in an amplicon that when sequenced with paired‐end reads has at least ~50 bp of overlapping sequence in the middle. For example, if running 2x300 bp paired‐end reads Illumina recommends having an insert size of 550 bp or smaller so that the bases sequenced at the end of each read overlap [16S Metagenomic Sequencing Library Preparation Guide, Illumina].

comment5 Page 7, Line 160-162, the meaning of this statement is that the data needs to be rarefied before performing alpha diversity analysis? Please rewrite this description to make it clearer.

response5

Thank you for your question. To improve clarity, we have revised the statement as follows:

Revised sentence (Page8, Lines 167-169)

"To adjust for uneven sequencing depth among the samples, the samples were rarefied to even depths of 10,000 before diversity analysis."

comment6 Page 7, Line 166, the “between” should be “among”.

response6

Thank you for pointing this out. We have revised it as follows:

Revised sentence (Page 8, Line 172):

"Correlations among microbiome, functional pathways, and orthologous groups were analyzed using Pearson correlation coefficients. All statistical analyses were performed using R version 4.3.2."

comment7 The “Materials and Methods” only has a small section on macrogenomes and some description could be added.

response7

Thank you for your valuable comment. In this study, we focused on 16S rRNA-based metagenomic analysis, rather than whole macrogenome sequencing. Our primary objective was to characterize the bacterial community structure and functional potential in airborne and soil environments using taxonomic profiling and predictive functional analysis (Tax4Fun).

We acknowledge that macrogenomic analysis could provide additional insights into the genetic potential of microbial communities. However, given the scope and objectives of this study, we opted for a 16S rRNA approach to efficiently compare bacterial and extracellular vesicle (BEV) compositions. Future studies may incorporate whole-genome sequencing to further explore microbial functional capacities in greater detail.

We have clarified this point in the Materials and Methods section to avoid potential misunderstandings.

Revised sentence (Page 7, Line 136-141):

“In this study, we employed 16S rRNA-based metagenomic analysis to characterize bacterial communities rather than whole macrogenome sequencing. This approach was chosen to efficiently compare bacterial and BEV compositions in airborne and soil environments. Bacterial genomic DNA targeting the V3–V4 hypervariable regions of the 16S rRNA gene was amplified with the 16s_V3_F and 16s_V4_R primers [34,35].”

comment8 Page 10, Line 232, significance here is not reflected in the data. Is there an additional significance test?

response8

Thank you for your valuable comment. We have reflected the significance in Figure 4.

comment9 Discussions about specific bacteria in air and soil are similar to the description of the results. Discussion is not a conclusion. In addition, “environmental factors strongly influence the microbiome” is based on a summary of other literatures, and there are no experiments in this manuscript. Is it possible to add some.

response9

Thank you for your valuable feedback. We have revised the Discussion to focus more on interpretation, comparisons with previous studies, and the significance of our findings in the context of environmental microbiology and human health. The revised sections are highlighted in red.

• Line 256-265

• Line 269-272

• Line 282-296

• Line 325-330

• Line 353-357

comment10 Similarly, the discussion of human health is not effectively integrated with the conclusions of this manuscript. It is not enough to summarize that “it is important to analyze the BEV microbiome for human health”. Please resolve this discrepancy.

response10

Thank you for your insightful comment. To better integrate the discussion on human health with the conclusions, we have revised sections where we discuss the implications of BEVs and bacterial communities in air and soil on health.

• Line 282-296

• Line 379-388

comment11 Page 13-14, Line 308-329, the discussion of aerosol in this place is a bit abrupt and some introduction could be added to the “Introduction” section.

response11

Thank you for your insightful comment. To improve the logical flow of the manuscript, we have revised the Introduction to provide better context for aerosol exposure.

Revised sentence (Page 3, Lines 52-59):

"Recently, interest in the microbiome has shifted from composition to functional roles in human health. Furthermore, public concern for ultrafine particles, including bioaerosols, has increased due to their ability to remain airborne for extended periods and penetrate deep into the respiratory tract. Among these bioaerosols, bacterial extracellular vesicles (BEVs) are emerging as key factors in airborne microbial exposure, potentially influencing immune responses and respiratory health [7, 8]. Understanding the differences between airborne and soil microbiomes, including BEVs, is critical for assessing their environmental and health implications."

comment12 Figure can be optimized a bit more, such as colors.

response12

Thank you for your valuable comment. We have optimized the figures by adjusting the colors for better visual clarity and enhancing the readability of significance levels through appropriate annotations.

comment13 The author should embellish some expressions. Please check the format and grammar of writing carefully.

response13

Thank you for your valuable feedback. We have carefully reviewed the manuscript for grammatical accuracy, formatting consistency, and overall clarity. In response to your suggestion, we conducted a thorough revision of the entire manuscript, making necessary corrections to improve readability and ensure precise scientific communication.

---

## [Decision Letter · Decision Letter 1]

27 Feb 2025

Examining the Bacterial Diversity including Extracellular Vesicles in Air and Soil: Implications for Human Health

PONE-D-24-49243R1

Dear Dr. Yang,

We’re pleased to inform you that your manuscript has been judged scientifically suitable for publication and will be formally accepted for publication once it meets all outstanding technical requirements.

Kind regards,

Jian Xu, Ph.D.

Academic Editor

PLOS ONE

Additional Editor Comments (optional):

Reviewers' comments:

Reviewer's Responses to Questions

**Comments to the Author**

1. If the authors have adequately addressed your comments raised in a previous round of review and you feel that this manuscript is now acceptable for publication, you may indicate that here to bypass the “Comments to the Author” section, enter your conflict of interest statement in the “Confidential to Editor” section, and submit your "Accept" recommendation.

Reviewer #1: All comments have been addressed

2. Is the manuscript technically sound, and do the data support the conclusions?

Reviewer #1: Yes

3. Has the statistical analysis been performed appropriately and rigorously? 

Reviewer #1: Yes

4. Have the authors made all data underlying the findings in their manuscript fully available?

Reviewer #1: Yes

5. Is the manuscript presented in an intelligible fashion and written in standard English?

Reviewer #1: (No Response)

6. Review Comments to the Author

Reviewer #1: Dear editor,

Thank you for inviting me to review the revision “Examining the Bacterial Diversity including Extracellular Vesicles in Air and Soil: Implications for Human Health”. I have carefully read the revised manuscript submitted by the authors and found that significant improvements have been made in response to the previous review comments.

7. PLOS authors have the option to publish the peer review history of their article (what does this mean? ). If published, this will include your full peer review and any attached files.

**Do you want your identity to be public for this peer review?** For information about this choice, including consent withdrawal, please see our Privacy Policy .

Reviewer #1: No

---

## [Editor Report · Acceptance letter]

PONE-D-24-49243R1

PLOS ONE

Dear Dr. Yang,

I'm pleased to inform you that your manuscript has been deemed suitable for publication in PLOS ONE. Congratulations! Your manuscript is now being handed over to our production team.

Kind regards,

on behalf of

Dr. Jian Xu

Academic Editor

PLOS ONE